# TGF-β Serum Levels in Diabetic Retinopathy Patients and the Role of Anti-VEGF Therapy

**DOI:** 10.3390/ijms21249558

**Published:** 2020-12-15

**Authors:** Vincenza Bonfiglio, Chiara Bianca Maria Platania, Francesca Lazzara, Federica Conti, Corrado Pizzo, Michele Reibaldi, Andrea Russo, Matteo Fallico, Elina Ortisi, Francesco Pignatelli, Antonio Longo, Teresio Avitabile, Filippo Drago, Claudio Bucolo

**Affiliations:** 1Department of Experimental Biomedicine and Clinical Neuroscience, Ophthalmology Section, University of Palermo, 90133 Palermo, Italy; enzabonfiglio@gmail.com; 2Department of Biomedical and Biotechnological Sciences, School of Medicine, University of Catania, 95123 Catania, Italy; chiara.platania@unict.it (C.B.M.P.); francesca.lazzara@unict.it (F.L.); conti.federica1@hotmail.it (F.C.); f.drago@unict.it (F.D.); 3Department of Ophthalmology, University of Catania, 95123 Catania, Italy; corradopizzo@hotmail.it (C.P.); andrearusso2000@hotmail.com (A.R.); matteofallico@hotmail.com (M.F.); elinaortisi@gmail.com (E.O.); ant-longo@libero.it (A.L.); t.avitabile@unict.it (T.A.); 4Department of Surgical Science, Eye Clinic, University of Torino, 10124 Torino, Italy; mreibaldi@libero.it; 5Department of Ophthalmology, SS Annunziata Hospital, 74121 Taranto, Italy; pignatelli.oculista@gmail.com; 6Center for Research in Ocular Pharmacology-CERFO, University of Catania, 95123 Catania, Italy

**Keywords:** diabetic retinopathy, serum biomarkers, anti-VEGFA, TGFβ

## Abstract

Transforming growth factor β1 (TGFβ1) is a proinflammatory cytokine that has been implicated in the pathogenesis of diabetic retinopathy (DR), particularly in the late phase of disease. The aim of the present study was to validate serum TGFβ1 as a diagnostic and prognostic biomarker of DR stages. Thirty-eight subjects were enrolled and, after diagnosis and evaluation of inclusion and exclusion criteria, were assigned to six groups: (1) healthy age-matched control, (2) diabetic without DR, (3) non-proliferative diabetic retinopathy (NPDR) naïve to treatment, (4) NPDR treated with intravitreal (IVT) aflibercept, (5) proliferative diabetic retinopathy (PDR) naïve to treatment and (6) PDR treated with IVT aflibercept. Serum levels of vascular endothelial growth factor A (VEGF-A), placental growth factor (PlGF) and TGFβ1 were measured by means of enzyme-linked immunosorbent assay (ELISA). Foveal macular thickness (FMT) in enrolled subjects was evaluated by means of structural-optical coherence tomography (S-OCT). VEGF-A serum levels decreased in NPDR and PDR patients treated with aflibercept, compared to naïve DR patients. PlGF serum levels were modulated only in aflibercept-treated NPDR patients. Particularly, TGFβ1 serum levels were predictive of disease progression from NPDR to PDR. A Multivariate ANOVA analysis (M-ANOVA) was also carried out to assess the effects of fixed factors on glycated hemoglobin (HbA1c) levels, TGFβ1, and diabetes duration. In conclusion, our data have strengthened the hypothesis that TGFβ1 would be a biomarker and pharmacological target of diabetic retinopathy.

## 1. Introduction

Diabetic retinopathy (DR) is a complication of diabetes mellitus, and it is generally defined as the microvascular retinal complication of diabetes [1,2]. DR is clinically classified as non-proliferative (NPDR) and proliferative (PDR). However, several substages have been identified in NPDR patients: early, moderate and severe NPDR. The latter is characterized by pervasive retinal hemorrhages and microvascular anomalies [3]. The risk to shift from NPDR to PDR is about 50%; in this perspective, the evaluation of prognosis and correct pharmacological management of NPDR would have a deep impact in the management of DR patients [4]. Diabetic macular edema (DME) is a main microvascular complication of PDR, although it can occur also in severe NPDR [5]. Furthermore, angiogenesis and inflammation are driving factors of DR and DME pathogenesis [6]. Therefore, current DR therapeutical approaches include intravitreal steroids [7] and anti-vascular endothelial growth factor (VEGF) agents, which are generally considered the first-line treatment [8]. 

Aflibercept is a human recombinant fusion protein that acts as a soluble decoy receptor for VEGF family members, including VEGF-A, VEGF-B and placental growth factor (PlGF) [9,10]. Aflibercept is approved with the following indications: neovascular age-related macular degeneration, macular edema following retinal vein occlusion, diabetic macular edema and diabetic retinopathy [11,12]. Furthermore, aflibercept exerts in vitro and in vivo anti-inflammatory action, modulating the phosphorylation of extracellular signal-regulated kinases (ERK) and decreasing retinal tumor necrosis factor alpha (TNF-α) release [13]. Besides angiogenesis and inflammation, retinal fibrosis has emerged as a detrimental factor in PDR pathogenesis [14,15]. TGFβ signaling pathway is strictly involved in fibrosis and the remodeling of the extracellular matrix [16,17]; and several reports highlighted that TGFβ can be implicated in the burden of PDR [18], promoting retinal fibrotic events [15]. Moreover, TGFβ pathways could promote angiogenesis, along with VEGF [19,20]; and TGFβ isoforms 1-2-3 were reported to induce VEGF expression [20,21,22]. 

Most of the retrieved studies reported that TGFβ-signaling activation is detrimental in DR and age-related macular degeneration (AMD) [20,21,22,23]. Additionally, some controversial recent data, generated from AMD models, suggested that TGFβ signaling activation, through TGFβ receptor 2 (TGFβR2), would protect the retina from neuroinflammation and apoptosis, regulating microglia activation and the expression of retinal neurotrophic factors [24,25,26]. Indeed, the mechanisms underlying TGFβ pathway activation and retinal neovascularization are complex; furthermore, related data are generally controversial [27]. It has been demonstrated that TGFβ signaling, through endoglin receptor, promoted subretinal fibro-neovascularization [28]. Shen et al. (2008) demonstrated that anti-VEGF treatment inhibited retinal TGFβ signaling, reducing p-smad3 levels and leading to decreased inflammation and retinal microglia activation [28]. Intriguingly, recent reports suggested that microRNAs, regulators of angiogenesis and the TGFβ signaling pathway, would be predictive biomarkers of early phase diabetic retinopathy [19,29,30], along with structural optical coherence tomography (S-OCT) assessment and other clinical outcomes and biomarkers [31]. 

Therefore, in this pilot study, we tested the hypothesis that the intravitreal injection of aflibercept in DR patients would influence not only the serum levels of VEGF-A and PlGF, but also TGFβ1. Furthermore, using several statistical analyses (C-statistics of receiver operating characteristics ROC curves and Multivariate-ANOVA), we analyzed the serum levels of VEGF-A and placental growth factor (PlGF), as well as TGFβ1, to validate novel biomarkers of DR classification and/or new pharmacological targets. 

## 2. Results

### 2.1. Study Subjects and Ophthalmic Evaluation

Thirty-eight subjects that fulfilled the eligibility criteria were included in the study (19 males, 19 females, mean age 70 ± 9) and assigned to six groups (Table 1). Participants’ demographics and pre- operative data are reported in Table 1. As regards healthy controls and diabetic, naïve NPDR and PDR patients, fasting venous blood sampling was carried out at the time of the inclusion in this study, after general ophthalmic evaluation and informed consent signature. NPDR and PDR patients that underwent aflibercept treatment were subjected to fasting venous sampling 7 days after intravitreal injection of aflibercept. 

### 2.2. Clinical Assessment

All diabetic subjects underwent OCT evaluation after study enrollment. Aflibercept-treated eyes (NPDR and PDR) underwent OCT evaluation also 7 days after intravitreal IVT injection (Figure 1). The average OCT foveal macular thickness (FMT) of diabetic patients, without signs of diabetic retinopathy, was 221 ± 15 µm. NPDR naïve eyes (no aflibercept IVT treatment) reported a significantly higher OCT macular thickness, 479 ± 45 (*p* < 0.05), compared to diabetic eyes without signs of diabetic retinopathy. PDR naïve patients (no aflibercept IVT treatment) had a significantly higher FMT, 558 ± 30 µm (*p* < 0.05), than values in diabetic and NPDR naïve patients. Either NPDR or PDR patients, after aflibercept treatment, reported a significant decrease of OCT macular thickness (236 ± 50 and 289 ± 60, respectively), compared to NPDR and PDR naïve FMT (479 ± 45 and 558 ± 30 µm, respectively), and to the respective baseline values (data not shown, 512 ± 55 NPDR pre-treatment, 735 ± 60 PDR pre-treatment). After aflibercept treatment, the OCT macular thickness of NPDR patients did not differ from PDR-treated eyes (Figure 1). Furthermore, NPDR and PDR patients, after aflibercept treatment, showed reduced intraretinal cysts. Particularly DR eyes, before treatment, showed an irregular layered structure with flattening of the foveal depression and the presence of large cystoid spaces. 

### 2.3. Serum Growth Factor Levels

Levels of pro-angiogenic factors VEGF-A and PlGF have been evaluated in the serum of the study subjects. VEGF-A serum levels (Figure 2) in diabetic patients, naïve NPDR and PDR subjects, were significantly higher than the levels detected in the serum of age-matched control subjects. The VEGF-A serum levels of NPDR and PDR patients, one week after intravitreal treatment with aflibercept, were significantly decreased when compared to diabetic, naïve NPDR and PDR patients.

Furthermore, we evaluated PlGF levels in the serum of enrolled subjects (Figure 3). PlGF serum levels were higher (*p* < 0.05) in diabetic patients compared to healthy control subjects. No differences were detected between NPDR and PDR naïve patients compared to either control or diabetic patients. One week after aflibercept intravitreal injection, placental growth factor (PlGF) levels in NPDR patients were significantly increased (*p* < 0.05) compared to control, diabetic with no DR signs and NPDR naïve patients. 

TGFβ1 serum levels (Figure 4) were significantly (*p* < 0.05) higher in the diabetic group, compared to control. Although not significant, TGFβ1 levels were higher in NPDR naïve patients, compared to diabetic patients without signs of DR. NPDR patients treated with aflibercept, 7 days after the last injection, showed a significant reduction in TGFβ1 levels, compared to naïve NPDR. PDR patients, either naïve or treated with aflibercept, showed significantly (*p* < 0.05) higher levels of serum TGFβ1, compared to other study groups. 

We aimed at validating TGFβ1 serum level as a specific and selective biomarker for DR patient stratification (Figure 5). C-statistics revealed that TGFβ1 levels predicted the classification of: (A). diabetic vs. healthy control patients (*p* < 0.0001, AUC = 0.94); (B). diabetic vs. naïve PDR (*p* < 0.0001, AUC = 0.89); (C). naïve NPDR vs. naïve PDR (*p* < 0.01, AUC = 0.81); (D). aflibercept-treated NPDR vs. aflibercept treated PDR patients (*p* < 0.0001, AUC = 0.93).

Serum TGFβ1 was not a valid biomarker for the differentiation of NPDR patients from diabetic patients without signs of DR. On the other hand, we validated serum TGFβ1 as a biomarker of DR progression from the NPDR to the PDR stage (Figure 5).

Furthermore, we carried out a Multivariate ANOVA in order to unveil the effects of fixed factors (independent variables) on dependent variables (diabetes duration, glycated hemoglobin HbA1c, TGFβ1, VEGFA, PlGF). In this perspective, we checked for the normal distribution of data and Pearson correlation coefficients. We tested with a M-ANOVA the effects of all independent variables only on the duration of diabetes, HbA1 and TGFβ1, because these dependent variables were normally distributed (Table 2) and correlated significantly (Table 3). 

The equality of the covariance matrix of dependent variables was satisfied, and the effects of independent variables (group, insulin treatment, glycemic control, gender) on HbA1c, TGFβ1 and the duration of diabetes were analyzed.

Group, glycemic control and their combinations (group * glycemic control; group * gender; glycemic control * gender) significantly (*p* < 0.05) affected the dependent variables, according to the multivariate analysis of variance (M-ANOVA) (Table 4 and Table 5).

Glycemic control significantly affected TGFβ1 serum levels in patients (Figure 6A). No statistically significant differences were observed between males and females, classified as “good control” or “poor control”, according to the provided medical reports (Figure 6A). Furthermore, we found that females showed differences in serum TGFβ1 levels, compared to males of the same group, although these differences were not always statistically significant (Figure 6B). Particularly, TGFβ1 levels in females were generally lower compared to males, in most of analyzed groups. On the contrary, females belonging to aflibercept-treated PDR group showed significant (*p* < 0.05) higher levels of TGFβ1, compared to males. This is because PDR females, treated with aflibercept, had poor glycemic control, and higher HbA1C (7.7 ± 1.1%) compared to males (6.7 ± 1.0%).

## 3. Discussion

Late diagnosis, duration of diabetes, poor glycemic control and lack of timely/appropriate treatment are the major causes of irreversible vision loss for DR patients [32]. Currently, steroid intravitreal implants/injections [7] and intravitreal injection of anti-VEGF agents are the approved pharmacological treatments of diabetic macular edema, in either non-proliferative (NPDR) or proliferative (PDR) diabetic retinopathy patients [33]. We hereby investigated in a pilot study the clinical outcome (FMT by OCT) and serum cytokines levels (VEGFA, PlGF and TGFβ1) in six groups of enrolled subjects: healthy controls (age-matched), diabetic without signs of DR, naïve and aflibercept-treated NPDR, naïve and aflibercept-treated PDR patients.

Seven days after the aflibercept IVT injection, FMT decreased significantly (*p* < 0.05) in both NPDR and PDR patients. This result is in accordance with current clinical practice results and with previous reports about anti-VEGF treatment outcomes in severe NPDR patients [34]. OCT foveal macular thickness was slightly higher, although not significantly, in PDR patients, compared to NPDR, after aflibercept treatment, according to previously published reports [35]. Then, we analyzed the VEGF-A and PlGF levels in the serum of all enrolled subjects. Specifically, VEGF-A serum levels were significantly (*p* < 0.05) higher in diabetic without signs of DR, NPDR and PDR naïve patients, compared to age-matched healthy subjects. One week after the intravitreal injection of aflibercept, VEGF-A levels decreased significantly in the serum of NPDR and PDR treated patients, compared to other groups. These data are in accordance with the effects of anti-VEGF intravitreal injections on serum VEGF-A, as reported in newborns with retinopathy of prematurity (ROP) [36] or in adults [37]. Moreover, we confirmed that VEGF-A in the serum of DR patients is not predictive of disease staging, as already reported in a previous study [38]. 

PlGF serum levels were not modified in diabetic patients, compared to controls. It is worthy of note that PlGF levels were significantly increased only in NPDR patients, one week after treatment with aflibercept, and no differences were reported in treated PDR patients compared to controls. This result is in accordance with previous published studies, both in the oncology and ophthalmology areas, describing the increase of serum PlGF as a counter-regulatory mechanism, due to VEGFR2 signaling inhibition by either VEGFR tyrosine kinase inhibitors or anti-VEGF agents [39,40]. Interestingly, the efficacy outcomes generated from the oncological clinical trial VELOUR were not influenced by either VEGF-A or PlGF serum levels, after intravenous injection of aflibercept [41]. Furthermore, in patients with neovascular AMD, PlGF serum levels were found to be increased 7 days after intravitreal injection of aflibercept [42], but authors did not associate the data with an analysis of the clinical outcome. Based on the data of our study, PlGF serum levels were neither predictive of DR staging nor of clinical outcomes. In fact, despite high PlGF serum levels in NPDR patients, the OCT showed a significant decrease of macular edema after aflibercept treatment. On the contrary, PlGF serum levels in PDR patients did not change 7 days after intravitreal injection of aflibercept. This result is probably related to uncontrolled retinal neovascularization in PDR patients, characterized by sustained VEGF signaling that, even if inhibited by an anti-VEGF, would mask any counter-regulatory expression of PlGF, that was observed in NPDR patients.

Serum levels of VEGFA and PlGF were not predictive of DR staging. On the other hand, TGFβ1 could be considered a sensitive, specific and validated biomarker of DR progression, according to our stratification analysis of subjects. We found significant C-statistics of ROC curves for TGFβ1 serum levels (healthy control vs. diabetic patients, diabetic vs. PDR, naïve NPDR vs. PDR, treated NPDR vs. PDR). On the contrary, on basis of the TGFβ1 levels, we were not able to differentiate naïve NPDR from diabetic patients without signs of DR. This could be due to limitations of our study, mainly accountable to the limited number of patients, and specifically to the heterogeneity of clinical characteristics of naïve NPDR compared to diabetic patients without signs of DR, which were not treated with insulin and showed low overall duration of diabetes and good glycemic control.

According to a previous study, we found that TGFβ1 serum levels of DR patients were higher than levels in diabetic group (without DR signs) and control subjects [43]. Furthermore, as regards the quantification of cytokines in sub-silicone oil fluid after vitrectomy, TGFβ1 levels were significantly higher (~3 fold) in patients with exacerbated PDR, compared to simple PDR (no re-proliferation of fibrotic membrane or vitreous hemorrhage) [44]. Moreover, TGFβ1 protein was found to be higher also in the aqueous humor of NPDR patients, compared to control subjects [45]. In particular, we observed that NPDR patients, after one week of treatment with aflibercept, showed significantly (*p* < 0.05) reduced serum levels of TGFβ1 and VEGF-A protein, while the PlGF protein amount was higher compared to that of naïve NPDR patients. These data could be indicative of aflibercept efficacy in NPDR patients. The analysis of the serum of PDR patients, treated with intravitreal injection of aflibercept, highlighted that only VEGFA levels were modified, along with the resolution of the macular edema. On the contrary, TGFβ1 levels were not significantly modified in PDR treated patients, compared to naïve PDR, possibly due to the clinical and demographic factors on the analyzed population. 

Based on this assumption, we carried out a multivariate ANOVA analysis (M-ANOVA), which showed that the diagnosis group, glycemic control and gender (independent variables) influenced TGFβ1, HbA1c and duration of diabetes (dependent variables). This analysis sheds light on the lack of statistically significant differences in TGFβ1 serum levels between naïve and aflibercept-treated PDR patients. In fact, the M-ANOVA analysis highlighted three outliers, bearing high TGFβ1 serum levels, in aflibercept-treated PDR group: i.e., females with poor glycemic control and higher HbA1c levels, compared to males. However, based on the present data, we cannot assert that gender influenced TGFβ1 serum levels and possibly a poor clinical outcome in aflibercept-treated PDR patients. We retrieved a recent pre-clinical report [46] that investigated the effects of sex difference on nephropathy in diabetic mice. This study showed higher renal TGFβ1 expression levels in female mice [46]. Sex hormones are reported to influence TGFβ1 [47], while in diabetes mellitus sex differences were found to be related to onset and duration of diabetes, glycemic control, puberty and menopause. In our study all females were in menopause age (see Table 1 reporting subject mean age), therefore we can conclude that gender effects retrieved with M-ANOVA in PDR-treated patients were outliers; i.e., females reporting poor glycemic control and higher HbA1c levels. However, a big longitudinal study would highlight gender effects on diabetic retinopathy.

Indeed, clinicians should strictly consider DR as a complication of diabetes, warranting a strict management of metabolic clinical outcomes. In this perspective, ophthalmologists should recommend to DR patients a correct management of glycemia and rigid compliance with diabetes therapy [48,49]. Furthermore, in DR management it would be useful to monitor not only the macular edema and retinal fundus, but also clinical laboratory parameters such as HbA1c, and possibly TGFβ1 serum levels. The main drawback of our study is the limited number of patients in each group, and a bigger longitudinal study would strengthen our data and the conclusions regarding the prognostic value of TGFβ1 in diabetic retinopathy.

In conclusion, TGFβ1 serum level can be considered a predictive biomarker of disease progression from NPDR to PDR, and it would likely be a secondary endpoint of anti-VEGF clinical efficacy, along with VEGF-A levels. Finally, TGFβ1 levels correlated with HbA1c levels and duration of diabetes. Indeed, these two variables should be taken into account by ophthalmologists during the clinical management of diabetic retinopathy.

## 4. Materials and Methods

### 4.1. Subjects

Subjects were enrolled at the Eye Clinic of the University of Catania. All subjects (19 males, 19 females, mean age 70 ± 9) (Table 1), including age-matched control subjects and diabetic patients without signs of DR (diabetic), diabetic patients with PDR and NPDR, read and signed the informed consent before enrollment. The study complied with the Declaration of Helsinki, and the protocol was approved by the Ethics Committee of the University of Catania (Project identification code #318). Inclusion criteria are hereby enlisted: age > 18 years, history of diabetes mellitus type 1 or 2 (diabetic patients). Only patients treated with aflibercept in one eye were included.

Exclusion criteria were macular edema not related to DR, recent ocular surgery (within 6 months), presence of epiretinal membranes/vitreomacular traction and incomplete medical records. Subjects were excluded in case of previous diagnosis of other proliferative vascular diseases, inflammatory diseases, and vitreous hemorrhages. Any previous intravitreal treatments, including both anti-VEGF and corticosteroids, were considered as exclusion criteria.

The diagnosis of non-proliferative and proliferative diabetic retinopathy was assessed by fundus examination using binocular ophthalmoscopy and fluorescein angiography.

Center-involving DME (central foveal macular thickness (FMT) > 300 μm) was assessed by Spectral Domain Optical Coherence Tomography (SD-OCT) (Optovue, Freemont, CA, USA; version 2017.1.0.151 AngioVue Phase 7 Software with PAR) using the retina map mode, which covered a 6.0 × 6.0 mm area centered at the fovea. 

Naïve NPDR and PDR patients with DME received aflibercept intravitreal injection (2 mg/0.05 mL—Eylea®, Bayer, Leverkusen, Germany) for the first time at the time of diagnosis. All injections were performed under sterile conditions in a surgical setting, after preparation of the conjunctiva using a 5% povidone–iodine solution, topical anesthetic, and positioning of the lid speculum. Ophthalmic clinical evaluation included fundus examination by binocular ophthalmoscopy, fluorescein angiography (FAG) and SD-OCT. All enrolled subjects underwent fasting venous blood sampling. In particular, blood samples from the NPDR and PDR aflibercept-treated group were collected 7 days after intravitreal injections. Serum samples were aliquoted and stored at −80 °C until. Serum samples from each subject were collected and masked with two randomly assigned digits (XY) (https://www.randomcodegenerator.com/), and the prefixes A-, B- and C- were assigned to each aliquot to be analyzed for VEGF-A, PlGF and TGFβ1 quantification, respectively. 

### 4.2. Enzyme-Linked Immunosorbent Assay (ELISA)

Serum levels of VEGF-A, PlGF and TGF-β1 were quantified by ELISA. Commercial ELISA kits: i. RAB0507 Millipore, Saint Louis, USA; ii. OKBB00242 Aviva systems biology, San Diego, CA, USA; iii. ADI-900-155 ENZO Life Science, Farmingdale, NY were used, respectively, for VEGF-A, PlGF and TGF-β1 quantification.

To quantify the VEGF-A levels, according to the manufacturer’s instructions, standards and samples were added into appropriate wells coated with anti-human VEGF-A, and the plate was incubated for 2.5 h at room temperature. After washing four times with the appropriate wash solution, the Biotinylated Detection Antibody was added to each well for 1 h at room temperature. Subsequently, the washing step has been repeated and a horseradish peroxidase (HRP)-streptavidine solution was added to each well for 45 min at room temperature. After washing again, 3,3′,5,5′-tetramethylbenzidine (TMB) substrate reagent was added for 30 min at room temperature in the dark. Lastly, Stop Solution has been added, and the absorbance at 450 nm was read immediately in a plate reader (VariosKan, Thermo Fisher Scientific, Waltham, MA, USA). 

In order to quantify PlGF, standards and samples were added in the anti-human PlGF pre-coated well plate and incubated at 37° for 90 min. After discarding the liquid in the wells, biotinylated anti-human PlGF antibody was added to each well, and the plate was incubated at 37° for 60 min. Subsequently the plate was washed three times with the specific wash buffer. According to the manufacturer’s instructions, Avidin-Biotin-Peroxidase Complex (ABC) was added into each well and incubated at 37° for 30 min. Then, the washing step was repeated five times, and TMB Color Developing Agent was added to each well for 15–25 min at 37° in the dark. Lastly, TMB Stop solution has been added, and the absorbance was read at 450 nm in a plate reader (VariosKan, Thermo Fisher Scientific, Waltham, MA, USA). 

For TGF-β1 detection, serum samples were activated by adding 2.5N acetic acid/10M urea. After 10 min of incubation at room temperature, the samples were neutralized with 2.7N NaOH/1M HEPES and Assay Buffer 13 was added, according to the manufacturer’s instruction. Activated samples and standards were added for 1 h at room temperature to wells coated with a human monoclonal antibody specific for TGF- β1. After washing four times with the specific Wash Buffer, a yellow solution of polyclonal antibody to TGF- β1 was added, and the plate was incubated for 2 h at room temperature. The plate was washed again to remove excess antibodies. A blue solution of HRP conjugate was added to each well and incubated for 30 min at room temperature. After washing, TMB substrate solution was added for 30 min at room temperature. Lastly, after adding Stop Solution, the optical density was read at 450 nm in a plate reader (VariosKan, Thermo Fisher Scientific, Waltham, MA, USA).

### 4.3. Statistical Analysis

OCT images and the demographic information of enrolled subjects were masked to investigators with random labels, assigned at the time of blood collection and serum sample labeling. Foveal macular thickness analysis, enzyme-linked immunosorbent assay (ELISA) quantification and statistical analysis were carried out by investigators unaware of the groups. The labels were unveiled after raw graph-design and statistical analysis. 

Statistical analyses were carried out with SPSS software, version 21.0 (SPSS Inc., Chicago, IL, USA), and GraphPad Prism (San Diego, CA, USA). The latter software was also used for graph design. Data were presented as mean ± standard deviation (SD). Parameters were checked for normal distribution, given a *p* < 0.05 of the Shapiro–Wilk test. A univariate ANOVA was carried out to test the effects of independent variables (age, gender, insulin treatment, glycemic control) on each single dependent variable (glycated hemoglobin HbA1c, duration of diabetes, TGFβ1, VEGFA, PlGF). Thereafter, given a significant F test (*p* < 0.05) and homogeneity of variance, a Tukey–Kramer post-hoc test was carried out for multiple comparison between subject groups. The significance level was set to *p* < 0.05. The diagnostic power of biomarkers was evaluated with ROC curves (C-statistics, AUC, confidence interval). Given the normal distribution of data, equality of covariance matrix and significant Pearson correlation for most of the dependent variables (duration of diabetes, HbA1c, TGFβ1), we carried out a multivariate ANOVA (M-ANOVA) to evaluate the effects of independent variables on all dependent variables.

## Figures and Tables

**Figure 1 ijms-21-09558-f001:**
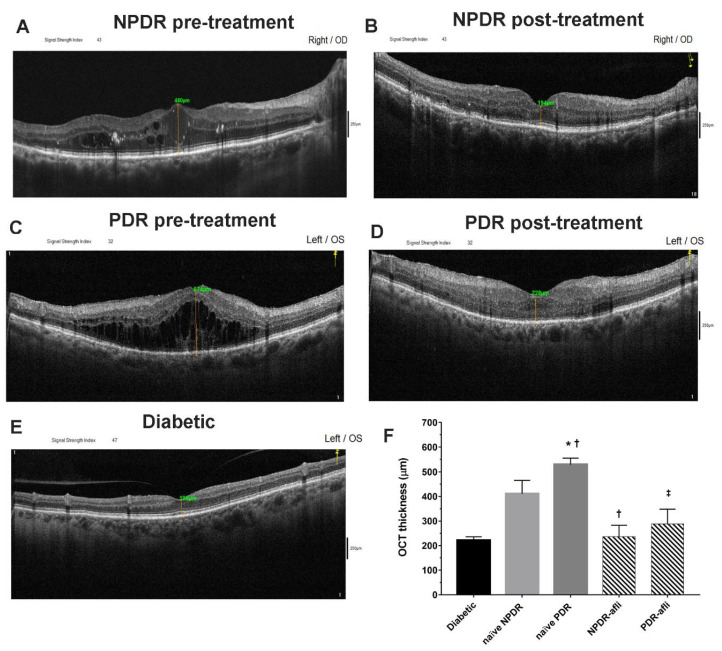
Representative optical coherence tomography (OCT) images of macular thickness. Aflibercept treatment decreased significantly (*p* < 0.05) foveal macular thickness in non-proliferative diabetic retinopathy (NPDR) and proliferative diabetic retinopathy (PDR) patients, compared to untreated naïve groups. Foveal macular thickness measurement in enrolled subjects beloging to the followig groups: NPDR before (**A**) and after (**B**) aflibercept treatment, PDR before (**C**) and after (**D**) aflibercept treatment, and OCT evaluation in diabetic patients without DR (**E**). Mean foveal macular thickness (**F**) μm ± S.D.; * *p* < 0.05 vs. diabetic; † *p* < 0.05 vs. NPDR naïve; ‡ vs. PDR naïve patients.

**Figure 2 ijms-21-09558-f002:**
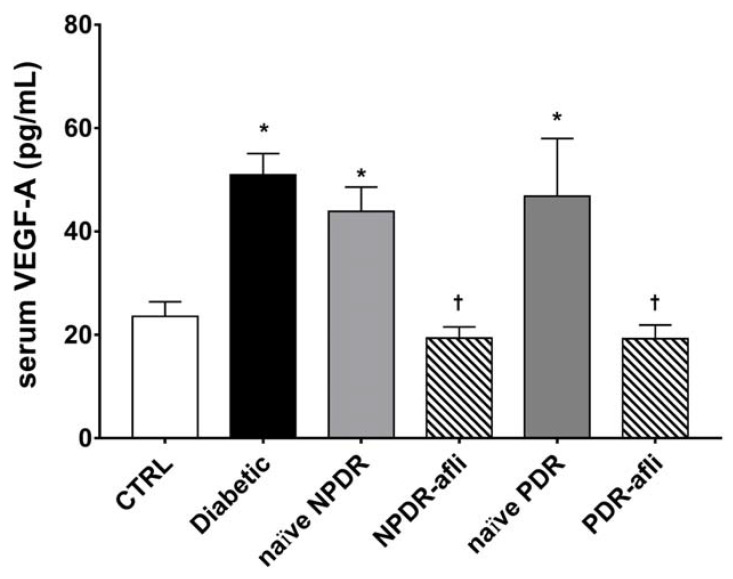
VEGF-A serum levels. After 7 days, aflibercept treatment significantly (*p* < 0.05) decreased VEGFA serum levels in NPDR and PDR patients, compared to diabetic patients without signs of DR, and compared to untreated naïve NPDR and PDR groups. * *p* < 0.05 vs. CTRL; † naïve vs. aflibercept (afli) treatment.

**Figure 3 ijms-21-09558-f003:**
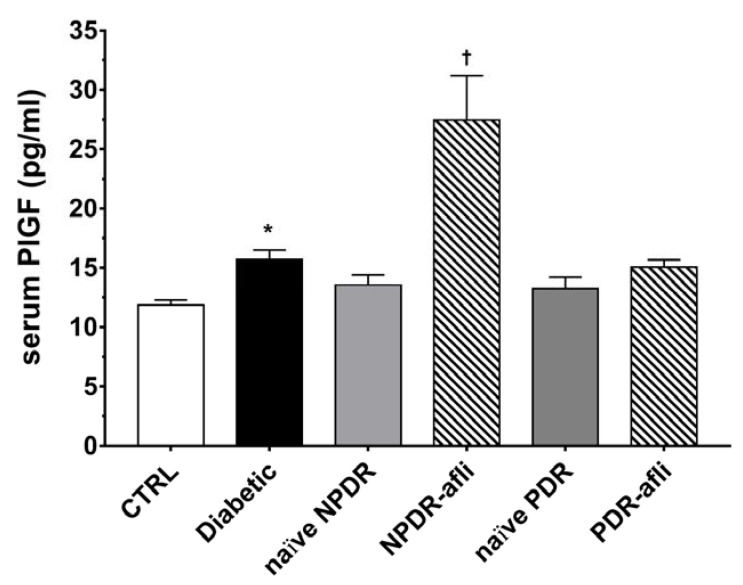
PlGF serum levels. After 7 days, aflibercept significantly (*p* < 0.05) increased PlGF serum levels only in NPDR treated patients, compared to other study subject groups. * *p* < 0.05 vs. CTRL; † naïve vs. aflibercept treatment.

**Figure 4 ijms-21-09558-f004:**
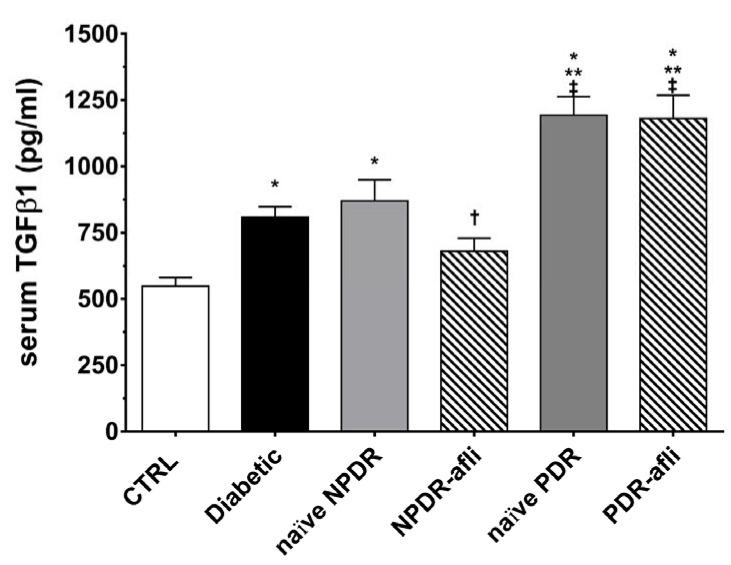
TGFβ1 serum levels. After 7 days, aflibercept significantly (*p* < 0.05) decreased TGFβ1 serum levels only in NPDR treated patients, compared to other study subject groups. * *p* < 0.05 vs. CTRL; ** *p* < 0.05 vs. diabetic patients without signs of DR; † *p* < 0.05 vs. NPDR naïve patients; ‡ *p* < 0.05 vs. NPDR either naïve or treated patients.

**Figure 5 ijms-21-09558-f005:**
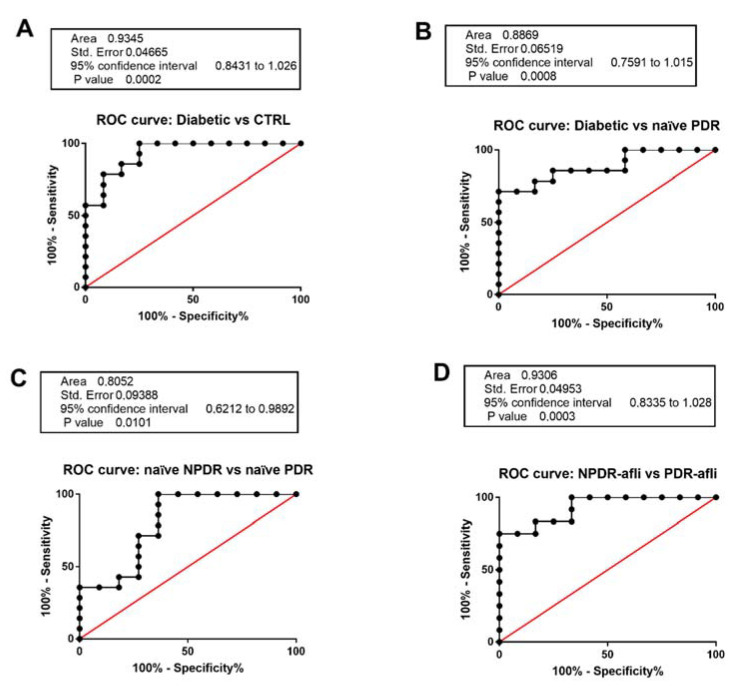
Receiving-operating characteristics curves for TGFβ1 serum levels. C-statistics validated TGFβ1 serum levels as a predictive biomarker of (**A**) diabetic patients without sign of DR (diabetic) compared to control healthy subjects; (**B**) diabetic compared to naïve PDR patients; (**C**) naïve NPDR compared to naïve PDR patients; (**D**) NPDR treated with aflibercept (afli) compared to PDR treated (afli) patients.

**Figure 6 ijms-21-09558-f006:**
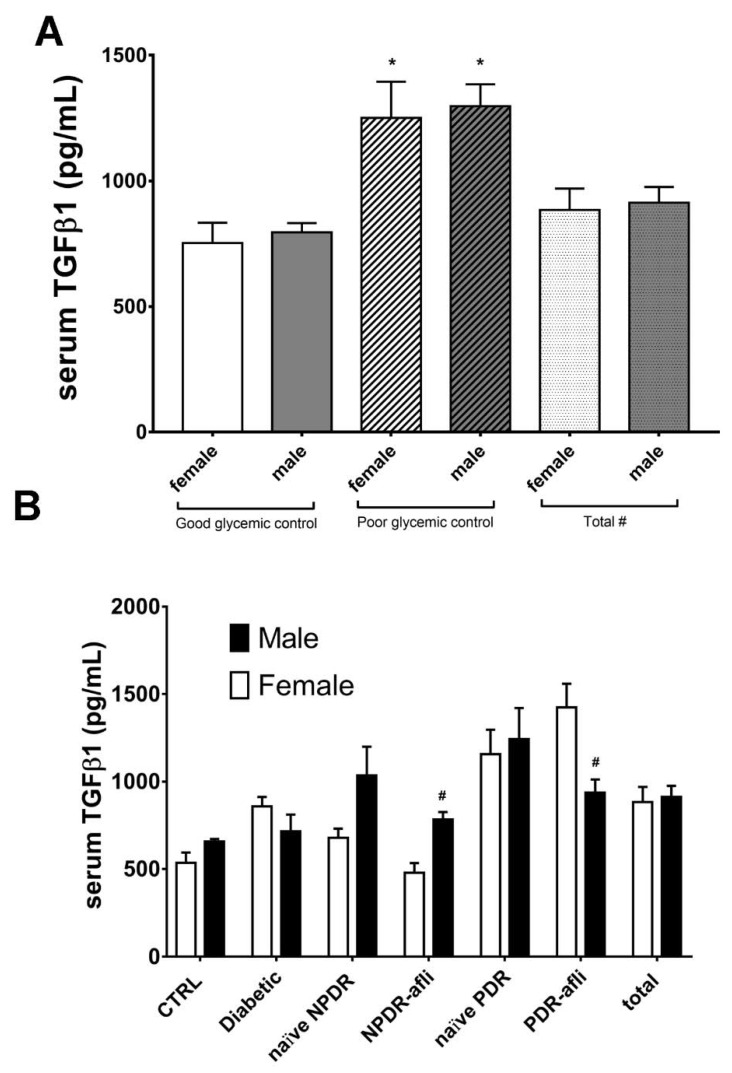
Glycemic control, group and gender effects on TGFβ1 serum levels. The M-ANOVA post-hoc analysis shed light on fixed factors (independent variables) effects on all analyzed dependent variables. A multivariate-ANOVA guided the stratification of TGFβ1 levels in terms of glycemic control and gender. (**A**) Both males and females, with reported poor glycemic control at enrollment, showed significantly (*p* < 0.05) higher levels of TGF1β, compared to other patients. The stratification of TGFβ1 on the basis of gender and subject group (**B**) showed that only PDR females treated with aflibercept had significantly (*p* < 0.05) higher levels of TGFβ1, compared to males of the same group. * *p* < 0.05 vs. “good glycemic control” group; # *p* < 0.05 males vs. females.

**Table 1 ijms-21-09558-t001:** Subject demographics.

	Gender (F; M)	HbA1c (%)	Diabetes Duration (Years)	Insulin Treatment (Y; N)	Subjects with Glycemic Control (Y; N)	Age (Years)
CTRL (N = 7)	(4; 3)	3.9 ± 0.9	NA	NA	NA	66 ± 14
Diabetic (N = 6)	(4; 2)	6.6 ± 0.3	5 ± 5	(0; 6)	(6; 0)	75 ± 10
NPDR naïve (N = 6)	(2; 4)	7 ± 1	19 ± 8	(4; 2)	(4; 2)	74 ± 6
NPDR aflibercept (N = 6)	(2; 4)	6.9 ± 0.5	20 ± 8	(5; 1)	(6; 0)	70 ± 7
PDR naïve (N = 7)	(4; 3)	7.3 ± 0.6	21 ± 6	(7; 0)	(2; 5)	70 ± 7
PDR aflibercept (N = 6)	(3; 3)	7 ± 1	18 ± 9	(3; 3)	(4; 2)	67 ± 8

F = females, M = males. Y = yes, N = no.

**Table 2 ijms-21-09558-t002:** Normality test.

	Kolmogorov–Smirnov			Shapiro–Wilk		
	Statistics	gf	Sign.	Statistics	gf	Sign.
Diabetes duration	0.163	38	0.014	0.926	38	0.017
HbA1c	0.190	38	0.002	0.914	38	0.007
TGFβ1	0.151	38	0.033	0.937	38	0.036
VEGF-A	0.154	38	0.028	0.794	38	0.000
PlGF	0.245	38	0.000	0.642	38	0.000

**Table 3 ijms-21-09558-t003:** Pearson correlation matrix of dependent variables. ** *p* < 0.01; * *p* < 0.05. Bold in order to further highlight statistically significant values.

		Duration	HbA1c	TGFβ1	VEGF-A	PlGF
Diabetes duration	Pearson coefficient	**1**	**0.595 ****	**0.335 ***	0.102	0.084
	Sign. (two tails)		**0.000**	**0.043**	0.547	0.622
	N	38	38	38	38	38
HbA1c	Pearson coefficient	**0.595 ****	**1**	**0.592 ****	0.271	0.163
	Sign. (two tails)	**0.000**		**0.000**	0.104	0.334
	N	38	38	38	38	38
TGFβ1	Pearson coefficient	**0.335 ***	**0.592 ****	**1**	0.003	−0.132
	Sign. (two tails)	**0.043**	**0.000**		0.984	0.436
	N	38	38	38	38	38
VEGF-A	Pearson coefficient	0.102	0.271	0.003	1	−0.156
	Sign. (two tails)	0.547	0.104	0.984		0.358
	N	38	38	38	38	38
PlGF	Pearson coefficient	0.084	0.163	−0.132	−0.156	1
	Sign. (two tails)	0.622	0.334	0.436	0.358	
	N	38	38	38	38	38

**Table 4 ijms-21-09558-t004:** Multivariate test of M-ANOVA. Bold in order to further highlight statistically significant values.

Effects	Wilks λ	F	*p*-Value
**group**	**0.037**	**6.836**	**0.0001**
insulin treatment	0.953	0.266	0.849
**glycemic control**	**0.358**	**0.573**	**0.001**
gender	0.869	0.805b	0.509
**Group * glycemic control**	**0.627**	**3.167**	**0.05**
**Group * gender**	**0.39**	**2.049**	**0.05**
**glycemic control * gender**	**0.464**	**6.152**	**0.006**

**Table 5 ijms-21-09558-t005:** Between-subjects effects of M-ANOVA. Bold in order to further highlight statistically significant values.

Source of Variation	Dependent Variable	F	*p*-Value
**group**	**HbA1c**	**9.624**	**0.0001**
	**TGFβ1**	**12.708**	**0.0001**
	Diabetes duration	2.077	0.116
Insulin treatment	HbA1c	0.35	0.562
	TGFβ1	0.272	0.608
	Diabetes duration	0.095	0.762
**Glycemic control**	**HbA1c**	**13.579**	**0.002**
	**TGFβ1**	**6.873**	**0.017**
	**Diabetes duration**	**4.582**	**0.046**
gender	HbA1c	0.486	0.494
	tgfbeta1	1.998	0.175
	Diabetes duration	0.062	0.806
**group * glycemic control**	**HbA1c**	**6.217**	**0.023**
	TGFβ1	0.112	0.742
	Diabetes duration	1.523	0.233
**group * gender**	HbA1c	0.337	0.799
	**TGFβ1**	**6.253**	**0.004**
	Diabetes duration	1.123	0.366
**Glycemic control * gender**	HbA1c	0.068	0.798
	**TGFβ1**	**15,571**	**0.001**
	Diabetes duration	2.478	0.133

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
