# Peer review of "TGF-β Serum Levels in Diabetic Retinopathy Patients and the Role of Anti-VEGF Therapy"

_ijms, 2020, doi:10.3390/ijms21249558_

Round 1
Reviewer 1 Report
This paper deals with an important topic, the evaluation of serum TGFb1 besides VEGF and PlGF at different stages of DR and after the treatment with IV aflibercept. The authors have found that in PDR patients, either naïve or treated with aflibercept, showed significantly (p<0.05) higher levels of serum TGFβ1, compared to other study groups.
The introduction is well written and the references are all pertinent to this topic, as well as extensive. The methodology is clearly described and the strength of the study is the masked evaluation of serum parameters and OCT findings.
The results are clearly described.Discussion extensively describes the results.
On a minor note:
- Given the limited number of included patients divided in different groups, the pilot nature of the study should be clearly stated both in the introduction as well as in the discussion.
- The conclusions are very interested. However, I would suggest to the authors to tone-down the significance as prognostic factor of TGFb1 from worsening of NPDR to PDR, as a longitudinal study with higher numbers would be needed for this statement.
- What is the rationale for the authors to evaluate separately TGFb1 in females vs males?
- Some typing errors throughout the manuscript should be corrected
Author Response
Rebuttal letter
Referee 1
This paper deals with an important topic, the evaluation of serum TGFb1 besides VEGF and PlGF at different stages of DR and after the treatment with IV aflibercept. The authors have found that in PDR patients, either naïve or treated with aflibercept, showed significantly (p<0.05) higher levels of serum TGFβ1, compared to other study groups.
The introduction is well written and the references are all pertinent to this topic, as well as extensive. The methodology is clearly described and the strength of the study is the masked evaluation of serum parameters and OCT findings.
The results are clearly described.Discussion extensively describes the results.
On a minor note:
- Given the limited number of included patients divided in different groups, the pilot nature of the study should be clearly stated both in the introduction as well as in the discussion.
Answer: Thank you. we now added this statement in the main section of the manuscript.
- The conclusions are very interested. However, I would suggest to the authors to tone-down the significance as prognostic factor of TGFb1 from worsening of NPDR to PDR, as a longitudinal study with higher numbers would be needed for this statement.
Answer: The referee is right, we now modified the conclusion accordingly.
- What is the rationale for the authors to evaluate separately TGFb1 in females vs males?
Answer: Thank you for your observation. We found a significant change in terms of TGFβ1 levels in NPDR patients treated with aflibercept, but not in PDR patients treated with aflibercept. Therefore, we carried out M-ANOVA post-hoc analysis in order to shed light on fixed factors (independent variables) on all analyzed dependent variables. The results showed in figure 6, come from Multivariate-ANOVA, which evidenced a combined effect of group*gender and glycemic control*gender on TGFβ1 levels (tables 4 and 5). Therefore, with SPSS we stratified TGFβ1 levels in terms of glycemic control and gender. Both males and females, with reported poor glycemic control at the time of enrollment, showed higher levels of TGF1β, compared to other subjects (Figure 6A). The stratification of TGFβ1 on basis of gender and subject group (Figure 6B) was also done, and only in PDR group (treated with aflibercept) females showed significant higher levels of TGFβ1, reporting also higher HbA1C (7.7±1.1 %) compared to males (6.7±1.0%). Since this is a pilot study we only concluded that glycemic control, and related HbA1C are key factors regulating TGFβ1, at least on the basis of our data. We discussed further this issue.
- Some typing errors throughout the manuscript should be corrected
Answer: We now checked and corrected the manuscript for typos.
Reviewer 2 Report
As a whole I consider the study here presented by Bonfiglio and colleagues scientifically sound, and without doubt it will be of interest for the readers of International Journal of Molecular Science. The results are interesting and properly discussed. However, some aspects could be improved.
Minor points
In the introduction section, the statement “is the secondary complication of diabetes” should be edited.
The figure 6 shows higher TGFb1 in female NPDR-afli group compared to male NPDR-afli. Authors should better discuss this issue. Is there any evidence in literature supporting this observation?
The legends of the figures need to be improved. At the current state they just give few details. Authors should describe the results to help the reader to understand the meaning of what the figure is showing, bearing in mind, that journal is intended to a broad reader audience and a well-written figure legends can ease the comprehension by the non-specialists.
Authors should review the labels of the figures, the groups are not reported consistently through the figures.
Author Response
Referee 2
As a whole I consider the study here presented by Bonfiglio and colleagues scientifically sound, and without doubt it will be of interest for the readers of International Journal of Molecular Science. The results are interesting and properly discussed. However, some aspects could be improved.
Minor points
In the introduction section, the statement “is the secondary complication of diabetes” should be edited.
Thank you, done
The figure 6 shows higher TGFb1 in female NPDR-afli group compared to male NPDR-afli. Authors should better discuss this issue. Is there any evidence in literature supporting this observation?
Answer: Thank you for your observation. We found a significant change in terms of TGFβ1 levels in NPDR patients treated with aflibercept, but not in PDR patients treated with aflibercept. Therefore, we carried out M-ANOVA post-hoc analysis in order to shed light on fixed factors (independent variables) on all analyzed dependent variables. The results showed in figure 6, come from Multivariate-ANOVA, which evidenced a combined effect of group*gender and glycemic control*gender on TGFβ1 levels (tables 4 and 5). Therefore, with SPSS we stratified TGFβ1 levels in terms of glycemic control and gender. Both males and females, with reported poor glycemic control at the time of enrollment, showed higher levels of TGF1β, compared to other subjects (Figure 6A). The stratification of TGFβ1 on basis of gender and subject group (Figure 6B) was also done, and only in PDR group (treated with aflibercept) females showed significant higher levels of TGFβ1, reporting also higher HbA1C (7.7±1.1 %) compared to males (6.7±1.0%). Since this is a pilot study we only concluded that glycemic control, and related HbA1C are key factors regulating TGFβ1, at least on the basis of our data. We discussed further this issue.
The legends of the figures need to be improved. At the current state they just give few details. Authors should describe the results to help the reader to understand the meaning of what the figure is showing, bearing in mind, that journal is intended to a broad reader audience and a well-written figure legends can ease the comprehension by the non-specialists. Authors should review the labels of the figures, the groups are not reported consistently through the figures.
Answer: Thank you for your observation, we now fixed this issue